# A Systematic Review of the Biological Effects of Cordycepin

**DOI:** 10.3390/molecules26195886

**Published:** 2021-09-28

**Authors:** Masar Radhi, Sadaf Ashraf, Steven Lawrence, Asta Arendt Tranholm, Peter Arthur David Wellham, Abdul Hafeez, Ammar Sabah Khamis, Robert Thomas, Daniel McWilliams, Cornelia Huiberdina de Moor

**Affiliations:** 1Pain Centre Versus Arthritis, University of Nottingham, Nottingham NG7 2RD, UK; masar.radhi1@nottingham.ac.uk (M.R.); asta.tranholm@nottingham.ac.uk (A.A.T.); dan.mcwilliams@nottingham.ac.uk (D.M.); 2School of Pharmacy, University of Nottingham, Nottingham NG7 2RD, UK; steven.lawrence1@nottingham.ac.uk (S.L.); peter.wellham@nottingham.ac.uk (P.A.D.W.); abdul.hafeez@nottingham.ac.uk (A.H.); ammar.khamis@nottingham.ac.uk (A.S.K.); 3Aberdeen Centre for Arthritis and Musculoskeletal Health, Institute of Medical Sciences, Aberdeen AB25 2ZD, UK; sadaf.ashraf@abdn.ac.uk; 4The Primrose Oncology Unit, Bedford Hospital NHS Trust, Bedford MK42 9DJ, UK; rt@cancernet.co.uk; 5Department of Oncology, Addenbrooke’s Cambridge University Hospitals NHS Trust, Cambridge CB2 0QQ, UK; 6NIHR Nottingham Biomedical Research Centre (BRC), Nottingham NG5 1PB, UK

**Keywords:** cordycepin, natural product, signal transduction, AKT, mTOR, AMPK, ERK, inflammation, cell viability, review

## Abstract

We conducted a systematic review of the literature on the effects of cordycepin on cell survival and proliferation, inflammation, signal transduction and animal models. A total of 1204 publications on cordycepin were found by the cut-off date of 1 February 2021. After application of the exclusion criteria, 791 papers remained. These were read and data on the chosen subjects were extracted. We found 192 papers on the effects of cordycepin on cell survival and proliferation and calculated a median inhibitory concentration (IC_50_) of 135 µM. Cordycepin consistently repressed cell migration (26 papers) and cellular inflammation (53 papers). Evaluation of 76 papers on signal transduction indicated consistently reduced PI3K/mTOR/AKT and ERK signalling and activation of AMPK. In contrast, the effects of cordycepin on the p38 and Jun kinases were variable, as were the effects on cell cycle arrest (53 papers), suggesting these are cell-specific responses. The examination of 150 animal studies indicated that purified cordycepin has many potential therapeutic effects, including the reduction of tumour growth (37 papers), repression of pain and inflammation (9 papers), protecting brain function (11 papers), improvement of respiratory and cardiac conditions (8 and 19 papers) and amelioration of metabolic disorders (8 papers). Nearly all these data are consistent with cordycepin mediating its therapeutic effects through activating AMPK, inhibiting PI3K/mTOR/AKT and repressing the inflammatory response. We conclude that cordycepin has excellent potential as a lead for drug development, especially for age-related diseases. In addition, we discuss the remaining issues around the mechanism of action, toxicity and biodistribution of cordycepin.

## 1. Introduction

Cordycepin is the adenosine analogue 3’-deoxyadenosine, which is isolated from the caterpillar fungus *Cordyceps militaris* [1]. The caterpillar fungi are a popular health food and traditional medicine in China [2]. This interest has led to a large increase in the numbers of publications on cordycepin as a bioactive substance and potential medicine, as we illustrate in this study. The literature is fast growing, but some papers are lacking in rigour and many papers contain similar data gathered in different systems that are less than convincing on their own. In most cases, only a small selection of the available similar data are cited, hiding the repetitive nature of much of the published work. To review all the available evidence on the biological activities of cordycepin, we, therefore, chose an approach akin to the systematic reviews developed for social and clinical research, examining all relevant papers and attempting to pool data where possible. We focussed on five topics for which a lot of evidence was available: cell survival and proliferation, cell migration, cellular inflammation, signal transduction and animal models of disease. In the discussion, we indicate remaining issues that must be addressed in order for cordycepin to progress towards a medical application.

## 2. Results

### 2.1. History of the Field

Cordycepin was first isolated from *Cordyceps militaris* and characterised in 1950 [1]. It was widely used as a tool to investigate RNA processing in the early days of molecular biology research, especially after it became commercially available in the early 1970s [3,4,5,6,7]. This use of the compound waned in the 1980s, when molecular biology started focussing on the study of individual genes, which had often just been cloned and sequenced. Small molecule inhibitors were becoming less popular and regarded as inadequately specific for most purposes. This led to a marked reduction in publications on cordycepin, with only five publications in our database for 1997 (Figure 1a). Since then, publications on cordycepin are on the rise again, with an especially steep increase in the last decade. It is clear this is driven by the interest in cordycepin as the active component of a traditional medicine, with the majority of publications coming from Asia, especially China (Figure 1b).

### 2.2. Cell Death, Survival and Division

Early in the study of cordycepin, it was observed that high doses can kill cells in culture and affect cell proliferation [8,9]. The idea that cordycepin may be a potential cancer treatment has led to many studies examining the doses at which cordycepin reduces cell numbers in culture. The vast majority of the data for this category is in vertebrate cells, and we concentrated exclusively on these papers for this section.

A total of 103 papers contained data on effect of cordycepin on cell numbers in vertebrate tissue culture, 65 of which attributed this cytotoxicity at least in part to the induction of apoptosis (see the Methods section). In eight cases, autophagy was reported in addition to apoptosis [10,11,12,13,14,15,16,17]. MCF-7 breast cancer cells were found to die by autophagy in one study and by apoptosis in another [18,19].

The 50% inhibitory concentration (IC_50_) is commonly used to compare the effects of cytotoxic drugs on cell lines. This number incorporates both effects on cell death and on proliferation during the incubation time. The IC_50_ concentrations for cordycepin treatment of various cell lines ranged from 15 µM to 2 mM, with the incubation time usually being 24 or 48 h. By collecting IC_50_ data for 126 experiments from 74 papers (see the Methods section), we found that the average IC_50_ is 194 µM, with a standard deviation of 250 µM, reflecting the very large variability. However, as can be seen in Figure 2, most of the IC_50_ data cluster around the median at 135 µM, with a small number of almost resistant cell lines skewing the data. A caveat of this survey is that if a cell type does not reach the IC_50_ within the dose range tried, it will not appear in our analysis, so these data are likely to have the most validity for cell types and under conditions that do exhibit cordycepin sensitivity.

Many papers report that different cell types indeed have distinct sensitivities to cordycepin [20,21,22,23,24]. In addition, we found that sometimes the same cell lines had widely different IC_50_ values in different studies; some of these may be due to differences in the method for determining viability [25,26], while others are less easy to explain [11,27]. One potential cause for the diversity is that differences in the serum used in culture medium could differentially affect the stability of cordycepin in these experiments (see the Discussion section on the pharmacokinetics of cordycepin).

In contrast, we found 45 studies reporting no or very low cytotoxic effects of cordycepin on cells in culture at concentrations at which a desirable bioactivity was evident (cited in the Methods section). The doses ranged from 40 nM to 500 µM and had a median of 50 µM. Moreover, 19 of these studies indicated that cordycepin can actively promote cell survival or prevent senescence under conditions such as ER stress, oxidative stress, radiation, etoposide exposure and other disease-related cellular stresses [28,29,30,31,32,33,34,35,36,37,38,39,40,41,42,43]. In four cases, the survival was linked to the induction of autophagy [36,37,38,42,44]. These data suggest that cordycepin is not exclusively a cytotoxic compound.

Cordycepin is known to affect the cell cycle in oocytes and early embryos, presumably through its well-characterised effects on the cytoplasmic polyadenylation of mRNAs encoding cell cycle regulators [45]. Indeed, we found 15 papers confirming these effects on vertebrate oocyte maturation and embryonic cell division [46,47,48,49,50,51,52,53,54,55,56,57,58,59,60,61].

The role of cytoplasmic polyadenylation in normal mitotic cell division is, however, less clear [60,62]. This is an important question, as microtubule disrupting agents, which are common in nature, affect cell cycle progression and are potent cancer drugs [63]. To determine if cordycepin has distinct effects on the mitotic cell cycle, we selected papers with flow cytometry data obtained with a fluorescent DNA stain. With this method, the phase of the cell cycle that is halted by treatment can be determined by counting cells that have unreplicated DNA (G1 or G0 phase) have undergone replication but not cell division (G2 or M phase, double the amount of DNA) or are undergoing DNA replication (S phase, intermediate amount of DNA). Classical cell cycle inhibitors cause accumulation of cells in one of these cell cycle stages. We examined 31 sets of FACS data, which matched our criteria from 20 papers for the effects of cordycepin on cell cycle progression (for details, see the Methods section). Cordycepin doses that affected the cell cycle ranged from 5 µM to 1.6 mM, with a median of 80 µM and an average of 136 µM. It is interesting that these doses are lower but similar to those observed for the IC_50_ (median 135 µM). However, as dose-response data were not widely available, it is not possible to be sufficiently quantitative for a firm conclusion. As can be seen in Figure 3, although more datasets showed arrest in G2/M, similar numbers of studies found arrest in other single stages (G0/G1: 8, S: 8, G2/M: 11) and four experiments showed accumulation in both S and G2/M phase. Indeed, in several cases, different cell cycle effects were described in a single paper and these differences were associated with cell type, dose or timing of cordycepin treatment [27,64,65,66]. It has also been reported that the effects of cordycepin are dependent on the cell cycle phase of the cell at the time of treatment [67]. We can conclude that cordycepin commonly affects cell division, but does not have the distinct stage-specific effects associated with microtubule disruptors or other cell cycle inhibitors.

### 2.3. Cell Migration

Cell migration plays a large role in normal tissue development and the function of the immune system, but it is also associated with the metastasis of cancer cells and the progression of inflammatory diseases. We surveyed the papers with data on cordycepin in vertebrates for effects of cordycepin for data on cell migration and found 27 such papers; 26 of these papers reported a repression of cell migration in a variety of cell types including cancer derived cell lines, macrophages, smooth muscle cells and endothelial cells at doses between 0.4 and 400 µM and with a median of 100 µM (see the Methods section). Only one paper reported no effect of cordycepin on cell migration in myeloid leukemic cells, but the relevant data were not included in this paper [8]. In eight cases, the reduction of migration was associated with a cordycepin-mediated repression of metalloproteinases, enzymes which can degrade extracellular matrix to allow cells migrate through tissues and can activate the TGFβ family of cytokines [68,69,70,71,72,73,74,75,76]. Just one paper reported a cordycepin-mediated increase in metalloproteinases [77]. Other frequently noted changes in connection with cell migration were a reduction in the active form of Focal Adhesion Kinase (FAK, PTK2), a key player in metastatic cancer [74,78,79,80,81], and upregulation of the epithelial marker E-cadherin, suggesting a reversal of the epithelial-mesenchymal transition [66,73,77]. Induction of NFĸB-mediated transcription by inflammatory signalling is associated with the induction of cell migration and is discussed in the section on inflammation. The combined data clearly demonstrate that cordycepin is an inhibitor of cell migration.

### 2.4. Effects on the Inflammatory Response

At the cellular level, the inflammatory response is a well-characterised set of gene expression programmes that is activated by signals indicating tissue damage or infection [82]. The induced inflammatory genes include cytokines (e.g., TNFα, IL1β and TGFβ), prostaglandin synthases (e.g., COX-2 and PTGES), nitric oxide synthase (iNOS) and genes involved in cell migration and tissue remodelling, such as the cell adhesion molecule VCAM-1, and metalloproteinases, such as MMP9 (see also above). Many cells are capable of an inflammatory response, but cells such as macrophages and microglia are most sensitive. These specialised cells respond to a larger number of stimuli and amplify the response in tissues. In a healthy situation, the inflammatory response resolves following removal of the stimulus and tissue repair. In contrast, chronic inflammation is involved in many disease processes, including cancer metastasis, the induction of chronic pain, fibrosis and neurodegeneration [83,84,85,86]. In addition, the appearance of chronic low-grade inflammation is a recognised feature of the aging process and age-related diseases [87]. Therefore, we decided to evaluate the evidence for an effect of cordycepin on the inflammatory response.

We found 38 papers which described effects of cordycepin on inflammatory gene expression (see the Methods section). Of these papers, 36 reported a reduction of inflammatory products by cordycepin. A single paper reported inhibition of iNOS but induction of TNFα [88] and only one paper reported induction of multiple inflammatory genes by cordycepin [89]. The overwhelming evidence, therefore, indicates that cordycepin has anti-inflammatory effects in tissue culture in many cell types.

TGFβ is a cytokine family which is normally involved in stem cell maintenance, wound healing and resolution of the inflammatory response [90]. Chronic expression of these peptides is associated with cancer and pathogenic tissue remodelling, for instance in osteoarthritis, chronic kidney disease, heart failure and idiopathic pulmonary fibrosis [83,91,92,93]. We found five papers that indicated that cordycepin reduces responses to TGFβ in cell culture [36,78,94,95,96] and none that reported no effect or repression. In combination with the repressive effect on the TGFβ activating metalloproteases discussed above, these data suggest that cordycepin can inhibit TGFβ activity by both reducing activation and blocking the cellular response.

NFĸB is a transcription factor with key roles in activating genes during inflammation and wound healing [82]. Activation of inflammatory signalling cascades leads to translocation of NFĸB from the cytoplasm to the nucleus and binding to DNA. We found 11 papers reporting a cordycepin-mediated reduction in the nuclear levels of NFĸB [28,43,97,98,99,100,101,102,103,104,105]. In contrast, two papers reported no changes in the nuclear localisation of NFĸB [72,106]. The total protein or mRNA levels of NFĸB subunits were found to be reduced in four studies [107,108,109,110]. DNA binding or chromatin association of NFĸB in nuclear extracts was reported to be reduced in one paper [70], but unchanged in three papers, despite clear repressed inflammatory gene expression [72,97,111]. Four papers found a reduction in the phosphorylation of NFĸB subunits [105,109,112,113]. Surprisingly, one paper reported that NFĸB binding was required for the activation of a cordycepin-induced gene, indicating that for some genes, its activity is retained [114]. The data suggest that effects of cordycepin on NFĸB-mediated transcription occur at multiple levels and are somewhat variable, probably depending on which combination of gene, cell type and stimulus is studied.

In addition, when we were examining animal studies (further discussed below) we found 28 papers that indicated that inflammatory processes were inhibited by cordycepin in a variety of animal models of disease [28,37,94,97,110,115,116,117,118,119,120,121,122,123,124,125,126,127,128,129,130,131,132,133,134,135]. Collectively, the literature indicates that cordycepin has robust anti-inflammatory activity in both cells and animals.

### 2.5. Effects on Signal Transduction Pathways

A key goal of this systematic review was to obtain a clearer picture of the effect of cordycepin on signal transduction. We collected the papers studying the effects of cordycepin on signal transduction. After screening of the content, 79 papers were retrieved for detailed assessment of the effect of purified cordycepin on specific signal transduction pathways in tissue culture (Figure 4). Our assembled data showed that particular pathways were better investigated than others. The PI3K/Akt, mTOR, AMPK and MAPK signalling cascades were well represented and are depicted in Figure 5, Figure 6 and Figure 7, respectively. Disappointingly, 50 articles had to be excluded as the precise phosphorylation sites of the proteins were not indicated, neither in the description of the antibody, nor in the text of the paper, leading to ambiguity on which site was being studied. In the remaining papers, PI3K/Akt/mTOR signalling was the most commonly studied pathway (14 articles, 50%) followed by MAPK (13 articles, 43.33%) and finally AMPK (9 articles, 30%). A diverse set of cell culture models was used, the most common being various cancer cell lines, immune cells and neuronal cells. For a detailed flow chart of the study selection process including reasons for exclusion, see Figure 4.

### 2.6. Effects on the PI3K/mTOR/Akt Pathway

The phosphatidylinositol 3-kinase (PI3K) and its downstream effectors mammalian target of rapamycin (mTOR) and protein kinase B (AKT) pathways are closely intertwined pathways that regulate biological processes such as metabolic balance, growth, differentiation, cell migration and angiogenesis. They play a key role in a variety of human diseases, such as cancer, type 2 diabetes mellitus and neurodegenerative diseases, and have been implicated in aging[136,137,138,139,140]. Figure 5 shows a diagram giving the key features of this pathway.

The effect of cordycepin on mTOR activity is commonly assessed by its effect on the phosphorylation of S6 kinase (S6K), which is a major mTOR downstream substrate. This phosphorylation results in an increase in S6K activity. Conversely, it has been shown that S6 kinase phosphorylates mTOR at Ser^2448^ [141], which leads to inactivation of mTOR, indicating a negative feedback loop [12,66,142,143,144,145,146]. Three of the reviewed studies found a repressive effect of cordycepin on mTOR phosphorylation at site Ser^2448^ [11,44,142]. In contrast, one paper suggested that cordycepin increases phosphorylation of mTOR at Ser^2448^ [38] (Figure 8a).

AKT phosphorylation is the most widely studied modification in cordycepin-treated cells. AKT is a family of serine/threonine protein kinases that plays a key role in cellular proliferation, apoptosis and migration. AKT kinases consist of three conserved domains: an N-terminal PH domain, a central kinase CAT domain and a C-terminal extension (EXT) containing a regulatory hydrophobic motif (HM). AKT kinases are phosphorylated by their activating kinases at a threonine residue in the activation T loop (T308 in AKT1, T309 in AKT2 and T305 in AKT3) and a Serine residue in the C-terminal HM (S473 in AKT1, S474 in AKT2 and 41 S472 in AKT3) [147,148]. AKT1 is phosphorylated at Thr^308^ by PDK1 during growth factor stimulation. For maximal activation, phosphorylation at Ser^473^ by mTORC-2 further increases its activity[147,148]. In reviewing the impact of cordycepin on the phosphorylated level of Akt, the Ser^473^ site was extensively studied (11 papers). Most studies (nine articles) demonstrated an inhibitory effect of cordycepin on Akt phosphorylation at Ser^473^ [11,101,103,142,144,149,150,151,152,153]. Only one study found no effect [145]. These data are summarised in Figure 8b. Furthermore, only two studies investigated the effect of cordycepin on Akt phosphorylation at site Thr^308^; one concluded that cordycepin has no effect on Akt activation at this site [145] and the other showed a repressive effect [154]. In addition, total Akt has been reported to be reduced [152,155]. Overall, the surveyed literature indicates that cordycepin represses Akt phosphorylation by mTOR at Ser^473^.

**Figure 8 molecules-26-05886-f008:**
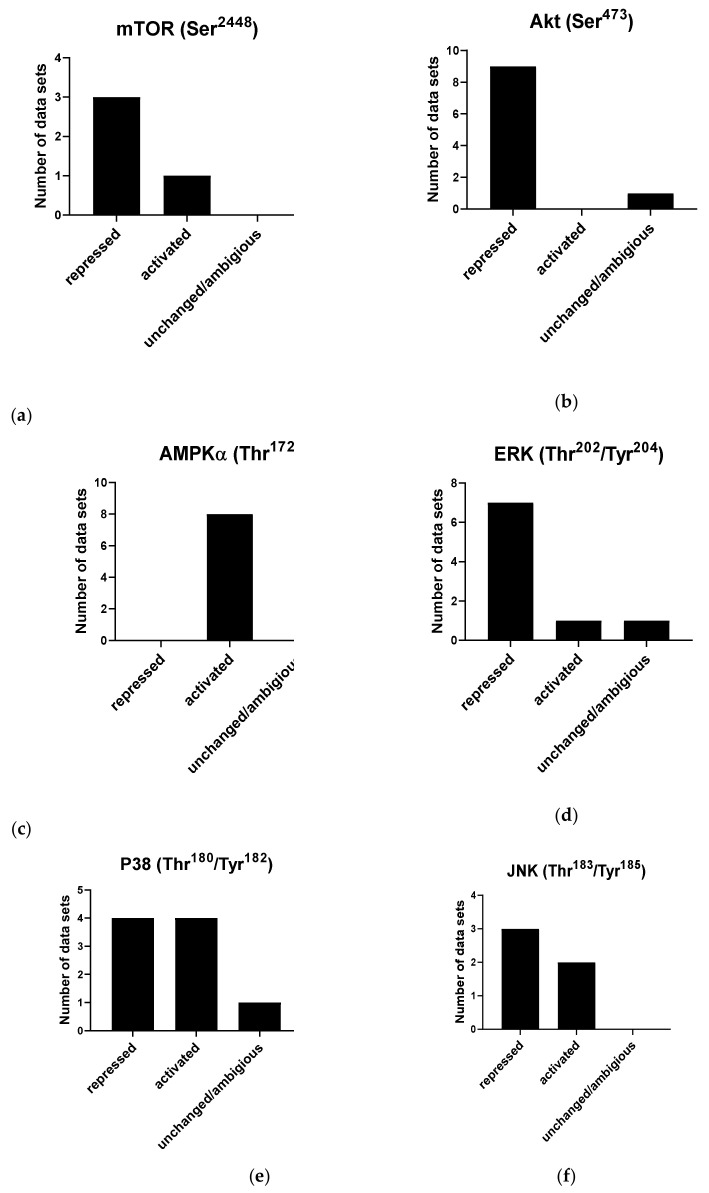
Effects of cordycepin on widely studied signal transduction pathways. Papers were selected as described in the Materials and Methods section and classified according to the effect observed on the indicated phosphorylation sites. (**a**) Effects of cordycepin on the Ser^2448^ phosphorylation site on mTOR; (**b**) Effects of cordycepin on the Ser^4738^ phosphorylation site on AKT; (**c**) Effects of cordycepin on the Thr^172^ phosphorylation site on AMPKα; (**d**) Effects of cordycepin on the Thr^202^/Tyr^204^ phosphorylation sites on ERK; (**e**) Effects of cordycepin on the Thr^180^/Tyr^182^ phosphorylation site on the p38 kinase; (**f**) Effects of cordycepin on the Thr^183^/Tyr^185^ phosphorylation site on the Jun kinase. The papers from which these data were extracted can be found in Table 1 in the Methods.

#### 2.6.1. Effects on AMPK Signalling

5’-adenosine monophosphate (AMP)-activated protein kinase (AMPK) is a highly conserved serine/threonine kinase. The AMPK kinase is a heterotrimeric protein complex containing a catalytic alpha subunit, an AMP-binding gamma subunit, and a scaffolding beta subunit. It is considered to be the master regulator of cellular energy homeostasis and is activated by upstream kinases, such as LKB1, CaMKKβ and Tak1 [156,157]. AMPK is allosterically regulated by the binding of AMP, or less commonly ADP, to its gamma subunit. This binding blocks the access of phosphatases and, thus, enhances the phosphorylation of Thr^172^ residue [158]. In the last decade, it has emerged that AMPK plays a central role in regulating a variety of metabolic and physiological processes. It is repressed in diverse medical conditions, such as overnutrition, inflammatory diseases, diabetes and cancer. Activation of AMPK also is the mechanism of action for the type II diabetes drug metformin. Therefore, activating AMPK could be a potential therapeutic target in treating these diseases [159,160,161].

The best-studied mechanism of AMPK activation is phosphorylation of the alpha subunit at Thr^172^, as depicted in Figure 6. AMPK activity is stimulated more than 100-fold by phosphorylation of Thr^172^ [162]. It has also been shown that phosphorylation at Ser^108^ of the beta subunit plays a key role in cell cycle regulation and promotion of pro-survival pathways in response to energy stress [163]. Although the Ser^108^ residue is auto-phosphorylated after prior phosphorylation at Thr^172^ residue, this residue is also the phosphorylation target of kinases other than AMPK such as ULK1 [163]. The literature search identified that a total of nine papers showed that cordycepin increases AMPK phosphorylation at Thr^172^ and Ser^108^, indicating activation in all cases [113,142,145,146,152,164,165,166,167]. Strikingly, we did not find a single article that suggested a repression or no effect on AMPK (Figure 8c). The literature unequivocally indicates that cordycepin treatment activates AMPK.

#### 2.6.2. Effects on MAPK Signalling

Various extracellular stimuli activate the mitogen-activated protein kinase (MAPK) cascade pathways [168]. Three distinct MAPKs have been widely investigated: p38 MAPK, c-Jun N-terminal kinases (JNK) and extracellular signal-regulated kinases (ERK), as depicted in Figure 7.

P38 proteins are a main subgroup of mitogen-activated protein kinases (MAPKs) that have been implicated in diverse biological processes, such as proliferation, differentiation, apoptosis and migration. Recent studies demonstrated that dysregulation of p38 has a key function in a wide variety of pathological conditions, such as solid tumours, arthritis and inflammation of the liver, kidney, brain and lung [169,170,171,172]. The activation of p38 is carried out by dual phosphorylation of their Thr–Gly–Tyr motif by MKK3 and MKK6 [173]. Nine studies investigating the effect of cordycepin on P38 met the eligibility criteria. Four studies showed a repressive effect [101,103,112,174], whereas four papers described that cordycepin activates P38 by increasing the phosphorylation at Thr^180^/Tyr^182^ [11,175,176,177]. A single study reported that cordycepin has no effect on P38 activation [178] (Figure 8d). Therefore, no clear trend for the effects of cordycepin treatment on p38 signalling could be discerned.

The c-Jun N-terminal kinases (JNKs) belong to the superfamily of the mitogen-activated protein kinase (MAPK) family involved in regulating eukaryotic cell reaction to a diverse range of cellular stress insults. They also coordinate essential physiological processes, including neuronal plasticity, immunological reactions, and embryonic development through their influence on gene expression, cytoskeletal protein dynamics, and cellular senescence [179,180]. Based on the search strategy employed, five articles investigating the effect of cordycepin on JNK met the inclusion criteria. Three of the studies showed a repressive effect of cordycepin on the phosphorylation of JNK at Thr^183^/Tyr^185^ [35,103,177], whilst the remaining two studies showed an activating effect on the same phosphorylation site [11,176] (Figure 8e). The effects of cordycepin treatment on JNK signalling are, therefore, ambiguous.

The extracellular signal-regulated kinases (ERK1 and ERK2) are evolutionarily conserved, highly regulated serine-threonine kinases that control cellular processes such as proliferation and differentiation. ERK plays a key role in development, and its upregulation is associated with the development and progression of diseases such as cancer. The ERK cascade is activated by signals such as growth factors, cytokines, viruses and G-protein-coupled receptor ligands [181]. The literature search yielded 38 publications related to effects of cordycepin on ERK signalling. Of these, only nine papers were included in the final analysis, as the majority did not indicate which phosphorylation sites were investigated. ERK1 and ERK2 are activated by dual phosphorylation by the upstream kinases MEK1/2 at a conserved threonine-glutamate-tyrosine (TEY) motif (Thr^202^ and Tyr^204^ in human ERK1, Thr^185^ and Tyr^187^ in human ERK2) [182]. Seven of the nine selected papers indicated an inhibitory effect of cordycepin on the phosphorylation of ERK at Thr^202^/Tyr^204^ [55,103,143,151,174,177,178]. A single article reported an activating effect of cordycepin on ERK [11], and another article reported no change [176] (Figure 8f). Overall, the literature suggests that cordycepin treatment usually inhibits the ERK pathway.

#### 2.6.3. Signal Transduction Effects in Animal Models

In reviewing the effect of cordycepin on signalling cascades in animal models (detailed below), eight articles were identified with data on signal transduction. Five of these met the eligibility criteria (see the Methods section). The outcomes were activation of AMPK indicated by increasing phosphorylation at Thr^172^ (three articles) and also inhibiting mTOR indicated by reduced the phosphorylation at Ser^2448^ (two articles), confirming the effects observed in tissue culture [35,142,164,183,184]. For details see Table 2.

### 2.7. Cordycepin Activity in Animal Models

Figure 9a reports the doses of purified cordycepin used in 131 animal studies in mg/kg. As can be seen, most studies use doses at 50 mg/kg or less, with 39 studies using 10 mg/kg or less. A few studies use very high doses. The routes of cordycepin administration to animal models were also evaluated. Figure 9b shows that 66 studies use intraperitoneal (IP) administration, followed by oral or intragastric administration (54), with only five studies using the intravenous route.

One of the first animal studies on the effects of cordycepin examined the effect of cordycepin on the growth of tumour cells in mouse ascites [185], but the follow-up of these studies took was long. However, we identified a total of 32 studies examining the effect of cordycepin on cancer animal models. Different cancers studied included those of the immune system, gastrointestinal tract, liver, breast and lung [11,18,23,24,64,66,74,79,80,107,108,134,142,143,186,187,188,189,190,191,192,193,194,195,196,197,198,199,200,201,202,203]. In most cases, cordycepin suppressed tumour growth, but it did not cause complete regression.

Concerning the cardiovascular system, 15 articles were found for a variety of animal models. Cordycepin was reported to have beneficial effects for cardiac hypertrophy [184], ischemia [122,183,204,205,206], dyslipidaemia [37,164,207,208] and other heart disease models [76,120,209,210,211].

Fourteen articles reported that cordycepin alleviates inflammation and pain in different animal models. Furthermore, cordycepin is described to have anti-inflammatory (reduced synovial inflammation) and analgesic effects in osteoarthritic animal models and in other conditions associated with pain and inflammation. Different mechanisms were proposed to potentially explain the effect of cordycepin on pain behaviour, such as inhibition of inflammatory signalling as well as direct effects on the primary afferent nociceptors [42,97,110,121,131,133,212,213,214,215,216,217,218,219].

Several studies strongly suggest that cordycepin can be used for treatment of trypanosomiasis, but that it needs to be stabilised to do so. Treatment with cordycepin, when combined with an inhibitor of adenosine deaminase, can prolong the survival of *T. evansi*-infected animal models and confer antiparasitic activity. In addition, the in vivo and in vitro activity of synthetic cordycepin derivatives has been studied in order to explain the structure–activity relationship [130,220,221,222,223,224,225,226,227,228,229,230,231].

We found 10 papers indicating that cordycepin can have protective effects on the brain in various models of brain dysfunction. For instance, it reduces the effects of chronic unpredictable mild stress in animal models and improves depression-like behaviour [119,232]. Some studies have also shown that cordycepin ameliorates learning and memory deficits in ischemic animal models [233,234,235,236] and sleep disturbances [237]. In addition, a substantial neuroprotective effect of cordycepin was noted in Machado-Joseph disease and Parkinson disease models [28,38,116].

Effects of cordycepin on animal models of respiratory disease were reported in eight papers. Multiple studies have shown an anti-asthmatic effect, which appears to be mediated by the inhibition of inflammation [123,124,128]. Other studies showed a protective effect of cordycepin in animal models of lung injury of inflammatory origin [117,118,129] and lung fibrosis [94].

The effect of cordycepin was also studied in animal models of metabolic diseases, including diabetes and obesity. Cordycepin has been shown to reduce body weight and change fat metabolism in animal models of obesity [166,238,239,240]. Studies in models of type II diabetes found that cordycepin treatment reduced plasma glucose level, hyperphagia and polydipsia. They also showed that hepatic glycogen content was dramatically increased, and oral glucose tolerance was enhanced after cordycepin treatment. Importantly, protective effects of cordycepin against diabetes-related kidney and spleen injury have also been reported. This effect is thought to be mediated through inhibiting cellular apoptosis and fibrosis, and inducing autophagy in diabetic nephropathy [241,242,243]. Thus, cordycepin appears to have beneficial effects on metabolic disorders and their related complications.

Eight studies indicated the effects of cordycepin on reproductive disorders [244,245,246,247,248,249,250,251]. These studies investigated the effect of cordycepin on testicular function, sexual behaviour and sperm production in rodent animal models.

## 3. Discussion

Our systematic review of the literature shows that cordycepin has significant effects in many animal models of disease and is an anti-inflammatory compound in both tissue culture experiments and animal models. The repression of inflammation is likely to be one of the key events in the therapeutic effects of cordycepin. While the mechanism by which cordycepin inhibits inflammation is not fully elucidated, it appears likely that the effects on signal transduction at least contribute. mTOR/AKT and AMPK are known to be involved in the regulation of inflammation and link metabolic changes to the inflammatory response in macrophages [252,253,254,255]. In addition, many of the disease models described attempt to mimic age-related conditions, such as arthritis, type II diabetes, heart disease and neurological damage, which are indeed linked to chronic inflammation [87]. Strikingly, other mTOR inhibitors (e.g., rapamycin/sirolimus) have been reported to increase lifespan (and therefore age) in low doses. Similarly, other AMPK activators (e.g., metformin) are well documented to improve metabolic health in aging individuals [140]. It is, therefore, not as unlikely as it first might appear that cordycepin has beneficial effects in so many apparently distinct, but age-related, conditions.

The cordycepin literature is overall of modest quality, as can be observed from common study flaws, such as the lack of information on which phosphorylation sites were examined, a frequent lack of primer sequence data and the few cases of image duplication (see the Methods section). This is likely due to the combination of meagre funding but high public interest for research into natural products, leading to large numbers of relatively low-budget studies and publications which are sometimes insufficiently critically reviewed. Nevertheless, this systematic review shows that the volume of studies indicating promising biological effects is now so large that the number of replicates is making up for any noise in the data. Our meta-analysis clearly shows that cordycepin has anti-proliferative and anti-inflammatory effects and that it activates AMPK, represses phosphorylation of AKT by mTOR and often reduces phosphorylation of ERK by MEK. While it can be argued that many of the animal models that have been used do not accurately replicate human disease, there can be no doubt that cordycepin has clear beneficial effects in many animals with a variety of disease-related symptoms.

A weakness of all systematic reviews is that conclusions can only be drawn on subjects that are widely researched and the choice of these subjects is dependent on the interests of the research community. It is, therefore, possible that the key biological effects or highest therapeutic potential are not summarised in this review. Nevertheless, the findings discussed above strongly indicate that cordycepin is an excellent lead compound for drug discovery, especially in cancer and age-related diseases. Indeed, we found 18 publications describing drug development projects tackling cordycepin modification and formulation [42,120,198,218,228,256,257,258,259,260,261,262,263,264,265,266,267].

A major outstanding issue that is hampering the development of cordycepin as a lead compound is the lack of a clearly identified cordycepin-binding target molecule and a mechanism of action that connects this binding with the therapeutic effects. Proposed binding targets include poly (A) polymerases, adenosine receptors, CDK2, PARP1, AKT, AMPK, FGFR2 and RuvB-like ATPase 2 (RUVBL2) [31,32,35,40,65,66,106,145,155,167,201,210,239,268,269,270,271,272,273,274,275,276,277,278]. A recent careful evaluation of AMPK as a cordycepin target concluded that although it is bound and activated by cordycepin monophosphate, AMPK activation is not responsible for the effects of cordycepin on cell survival [166]. However, the other proposed targets still remain to be fully characterised. It is noteworthy that we found four publications indicating that cordycepin has a repressive effect on an unidentified phosphorylation site of PI3K, suggesting the effects of cordycepin on mTOR may be caused by changes in upstream signalling events [26,239,279,280]. As cordycepin is a product of evolution, it is also possible that it has multiple targets that synergise to cause the biological effects [281]. A full discussion of the outstanding issues on the target identification of cordycepin is outside the scope of this review, but the large variety of systems and the similarities of the response discussed here indicate that the main target(s) of cordycepin cannot be very cell- or tissue-specific.

Several studies showed that cordycepin is efficiently converted to cordycepin triphosphate and trapped in cells, leading to accumulation. The inhibition of import and phosphorylation of cordycepin has been shown to reduce its effects, suggesting that intracellular and phosphorylated cordycepin is indeed at least one of the active metabolites of cordycepin [35,95,106,144,152,273,274,282,283,284,285]. As indicated in our results, cordycepin has shown biological activity in animal models when administrated intraperitoneally, intravenously or orally. In blood or tissue culture media, cordycepin is rapidly deaminated by adenosine deaminase, forming 3’ deoxyinosine [165,228,286,287,288,289]. After oral administration, even sensitive assays cannot detect any cordycepin in the circulation, causing some doubts as to the active metabolite of cordycepin. However, we recently showed that 3’ deoxyinosine can be converted into cordycepin triphosphate in at least some cell types [286]. This raises the possibility that cordycepin specifically targets particular tissues in the whole organism, not because of a tissue-specific molecular target, but because of tissue-specific conversion of 3’ deoxyinosine to cordycepin triphosphate. In addition, by circulating as 3’ deoxyinosine, cordycepin may be avoiding toxic effects, which can be caused by the accumulation of adenosine-like compounds [290]. It is, therefore, very important for further drug development that biodistribution studies of cordycepin, 3’ deoxyinosine and cordycepin triphosphate are performed to resolve this issue.

Another complicating factor in cordycepin research is the lack of commercially available highly purified and/or synthetic preparations. The most widely used preparation from Sigma (now Merck) is isolated from *Cordyceps militaris* and only 98% pure. As cordycepin is generally used in micromolar quantities, it is possible that there are contaminants active in the nanomolar range that contribute to the biological effects of cordycepin. The fact that similar effects have been observed over time and with different suppliers suggests that, if there are important bioactive contaminants, these should be very consistently present. Thankfully, in a few cases, purer and synthetic cordycepin preparations have been shown to have similar effects to those observed with the standard preparations [70,239,291]. It would help the field significantly if purer cordycepin preparations became commercially available as analytical standards and for the comparison of activities.

The toxicity of cordycepin in animals has been reported to be low in the absence of adenosine deaminase inhibitors [193,207,231,292], but we did not find any publications with dose escalation studies of cordycepin to several fold the therapeutic dose for intravenous or oral administration. For intravenous doses, this is understandable, as they are limited by the solubility of cordycepin in simple formulations [257]. Three studies with cordycepin administered intraperitoneally have yielded somewhat conflicting results, with one study claiming no adverse effects at 900 mg/kg and another reporting 50% lethality at 400 mg/kg and significant deaths after 3 days of 150 mg/kg daily [185,293]. A third study indicated that three of seven animals suffered weight loss, convulsions and death after a 3.6 g/kg intraperitoneal dose, while the other four survived [243]. Ames tests suggest that cordycepin has very weak or no mutagenic activity [193,294]. Moreover, chronic oral administration of lower doses of cordycepin appears to improve rather than decrease hepatic health [127,295]. Given the effects of cordycepin on the mTOR pathway, a remaining worry is that it might suppress the immune system, as does the mTOR inhibitor rapamycin (sirolimus). On the other hand, it is worth noting that low doses of mTOR inhibitors can improve immune responses in elderly patients [296]. Similarly, the inhibition of growth factor signalling could affect wound healing, although so far, cordycepin appears to promote healing [145]. Another concern is the documented effects of cordycepin on the meiotic cell cycle and early embryogenesis (see above) may affect especially female fertility. Full dose escalation experiments and careful study of the long-term effects of therapeutic doses with an emphasis on immunity, wound healing and fertility will be important to assess the safety of cordycepin.

Promising effects in animal models do not always translate into good medicines for human patients, and we will have to await the reports of clinical trials to know if cordycepin and its derivatives are also bioactive in people (e.g., NCT00003005, NCT00709215, NCT03829254, ChiCTR-INR-17014074). A trial of cordycepin in combination with the adenosine deaminase inhibitor pentostatin for acute lymphocytic leukaemia issued a preliminary account in 2000 [194], but unfortunately, this study has still not issued a final report. Promisingly, a trial of a partially purified cordycepin preparation for patients with chronic kidney disease has reported improvements in kidney function [109].

In conclusion, cordycepin has clear biological effects in a large number of animal models of disease. It has inhibitory effects on the PI3K/AKT/mTOR signalling pathway and activates AMPK. Most therapeutic effects of cordycepin are consistent with them being mediated by these effects on signal transduction. Moreover, the wide range of therapeutic effects reported in animal models is similar to other AMPK activators and mTOR inhibitors. Remaining challenges are the obscure mechanism of action of cordycepin, the lack of commercial availability of high purity cordycepin and the incomplete understanding of cordycepin biodistribution. Nevertheless, cordycepin appears to be an excellent drug lead for many common diseases and deserves to be further investigated.

## 4. Materials and Methods

### 4.1. Scoping of the Review

Only papers recorded in PubMed were considered. A PubMed search on “cordycepin” found 1167 publications entered in PubMed by 1 February 2021. We found 37 additional papers on the subject through references in other papers and by examining the abstracts for a PubMed search for “3’-deoxyadenosine NOT cordycepin” for relevant papers, yielding 1204 papers for initial examination.

We removed 39 articles not written in English from consideration. For the remaining papers, abstracts were then read to exclude publications solely containing data on cordycepin preparation, fungal production of cordycepin, crude *Cordyceps* extracts and chemical synthesis of cordycepin. We also excluded papers only containing data on chemical derivatives of cordycepin, with the exception of studies of the intracellular metabolite cordycepin triphosphate and animal studies. These filters removed 204 papers.

A total of 33 previous reviews not containing original data were also removed from consideration. Two corrections on included papers were removed from the database. The remaining publications were collated in an excel spreadsheet. Full articles were obtained from online sources, the University of Nottingham library and through the British Library. We failed to obtain full text copies for eight papers. Data on the biological effects of cordycepin were extracted from each paper by a member of the team using a spreadsheet.

A total of 910 papers potentially containing data on the biological effects of purified cordycepin remained at this stage.

### 4.2. Selecting Sources of Data

After initial reading, we classified these papers according to the system studied (vertebrate, arthropod, other animal, fungus, plant, protists, bacteria, Archaea and cell-free systems only). Viruses were classified with their host. If more than one system was studied, the earlier listed system (as above) was entered. Notes were entered on the characteristics and species studied and themes were extracted for further analysis.

We decided to exclude the papers on plants, as cordycepin is primarily used as a transcription inhibitor to elucidate contributions of this process to biological changes (123 papers). Some differences to other transcription inhibitors have been noted in seed germination, but the nature of this difference is unclear [297].

A total of 791 other papers remained containing data on effects of cordycepin (for a full list, see Appendix A). The organisms studied were vertebrates (655 papers), other animals (14 papers), fungi (31 papers), protists (30 papers) and cell-free systems (31 papers). Despite the fact that the most likely natural target of cordycepin is insects, only 13 papers tackled effects on arthropods. One paper on Archaea and 16 papers on bacteria were found. In most of this review, we concentrated on the effects on vertebrates.

For the third stage, papers describing vertebrate research were read in full to find data relating to our selected subjects. For each subject, the numbers of papers found are indicated in the relevant section. For tissue culture experiments, notes were entered in a standardised Excel spreadsheet under the following standardised headings: dose, time, cell survival, cell proliferation, cell migration, signal transduction (PI3K/mTOR/AKT, MEK/ERK, p38 and JunK, others), gene expression (inflammatory and other) and other effects. For papers describing animal experiments, notes were added on the following subjects: animal model, dose, administration route, inflammation, other immune system effects, bone and cartilage, tissue remodelling and fibrosis, tumour growth, neuronal effects, metabolic effects and other effects.

### 4.3. Quantifications

#### 4.3.1. Publication Year

The original 1204 papers were used to explore the publication year of cordycepin papers. Publication year was used as stated by PubMed.

#### 4.3.2. Geographic Origin

First affiliation of the last author was determined from the PubMed entry, or if absent, by consulting the paper if it was available to us. The continent of the affiliation was entered. If the last author had affiliations in more than one continent, the first stated was used. We determined the origin for 932 papers after excluding papers with unclear affiliation.

#### 4.3.3. Cordycepin Concentrations and Cell Viability in Mammalian Tissue Culture

We found 196 papers containing data on the effects of cordycepin on cell viability, death or proliferation and these were examined further for effects on cell number, cell survival and cell cycle.

A total of 103 papers contained data on the effects of cordycepin on cell numbers in vertebrate tissue culture. Data for modified cordycepin, cordycepin with other compounds such as pentostatin, cordycepin in protective or slow-release formulations or for cells in 3D culture were excluded. Of these 103, 74 papers included cell viability experiments which used colorimetric assays with tetrazolium salts (e.g., MTT: 3-(4,5-dimethylthiazol-2-yl)-2,5-diphenyltetrazolium bromide) or employed a live/dead cell counting method, e.g., trypan blue or propidium iodide, for which the 50% inhibitory concentration (IC_50_) for cells in conventional (2D) culture was determined or could be estimated from the data for the 24 h or 48 h timepoint for one or more cell lines (48 h was selected if both were available). These studies were used for the graph in Figure 2 [11,27,66,68,70,73,81,88,99,105,107,114,117,142,143,146,149,150,151,165,178,186,192,195,196,200,202,203,256,259,264,265,267,276,279,280,298,299,300,301,302,303,304,305,306,307,308,309,310,311,312,313,314,315,316,317,318,319,320]. A total of 65 papers were found to report cell death by apoptosis [10,27,65,66,73,77,88,99,107,142,143,146,150,151,155,174,178,186,189,192,196,198,200,201,202,203,264,276,279,280,299,300,302,303,304,305,307,308,309,310,311,314,319,320,321,322,323,324,325]. We did not filter these papers by the assay or by timing.

We noted 45 papers claiming that the cytotoxic effect of cordycepin at bioactive concentrations was low. We noted the highest concentration which promoted cell survival or did not cause statistically significant cell death in these papers and determined the median and the range [28,29,30,32,33,34,35,36,37,38,39,40,41,42,43,44,69,70,80,95,98,102,105,106,117,142,145,164,184,240,270,301,315,317,326,327,328,329,330,331,332,333,334,335,336].

Papers with notes on cell proliferation were examined for descriptions of the effect of cordycepin on cell cycle in vertebrate cells. A total of 58 papers containing data on cell cycle effects were found. We identified 20 papers with data generated by DNA staining and fluorescent automated cell counting that met our criteria [24,27,65,66,73,80,132,151,176,198,202,280,314,316,317,324,337,338]. We recorded cordycepin concentration and the effects on the distribution between G1/G0, S and G2/M for each cell line as increased, decreased or no significant change and the arrested stage was classed on the basis of these changes. Papers which did not separate the three cell cycle phases or which neglected to show replicated quantified data with statistical analysis were excluded. Data for modified cordycepin or cordycepin in combination with adenosine deaminase inhibitors were also excluded.

#### 4.3.4. Cell Migration

We found data on the effect of unmodified cordycepin in solution on the migration of vertebrate cells in 28 papers. Notes were entered on these experiments, including on effects on known regulators of cell migration. The lowest dose reported to give effects in scratch or transwell assays for each cell type was obtained from 19 papers [23,66,68,69,70,71,72,73,74,76,79,80,81,88,99,196,210,322,339]. Data in which cordycepin was combined with adenosine deaminase inhibitors, siRNAs or administered in a specific formulation (e.g., nanoparticles) were excluded from this analysis.

#### 4.3.5. Cellular Inflammation

Papers selected for analysis were examined for data of effects of cordycepin on cytokines, inflammation or regulation of inflammatory genes in tissue culture. We found 54 papers with data on these subjects [28,31,36,43,69,70,71,72,76,78,88,89,94,96,97,98,99,100,101,102,103,104,105,106,107,108,109,110,111,112,113,114,117,133,270,274,286,326,328,330,332,340,341,342,343,344,345,346,347,348,349,350,351,352]. We further examined data for nine genes: the cytokines Tumour Necrosis Factor α (TNFα), Interleukin 1β (IL1β) and Transforming Growth Factor β (TGFβ), the prostaglandin synthases Prostaglandin Synthase 2 (PTGS2, also known as cyclooxygenase 2-COX-2) and prostaglandin E synthase (PTGES), the inducible nitric oxide synthase NOS2 (also known as iNOS), the Vascular Adhesion Molecule 1 (VCAM1) and the metalloproteinases MMP-3 and MMP-9. Cordycepin-induced changes in the gene products were noted as increased, unchanged or decreased. We found 38 papers describing such changes [28,69,70,71,72,76,88,89,96,97,98,99,100,101,102,103,104,106,108,109,110,112,113,117,270,286,326,330,340,341,342,343,344,345,346,347,348,349]. In addition, notes were gathered on the effects on the nuclear localisation, level, phosphorylation and DNA binding of NFĸB in the 54 papers found for this subject.

#### 4.3.6. Signal Transduction

We found 76 papers on the effect of cordycepin on cellular signalling. Papers with data about PI3K/Akt/mTOR, AMPK and MAPK signalling cascades were examined for indicating the precise phosphorylation site of the studied kinases. A total of 47 papers were excluded because we could not find or infer which phosphorylation site was studied (i.e., no mention of the site or the antibody catalogue number). A total of 29 articles were finally included in the systematic review to study the effect of cordycepin on the following kinases: mTOR [11,38,44,142], Akt [11,101,103,142,144,145,149,151,152,153,155], AMPK [113,142,145,146,152,164,166,167], ERK [11,55,103,143,151,174,176,177,178], P38 [11,101,103,112,174,175,176,177,178] and JNK [11,35,103,176,177] (Table 1). The effects were classed as repressed, activated or unchanged/ambiguous (e.g., not statistically significant or conflicting results).

**Table 1 molecules-26-05886-t001:** Overview of the included publications on signal transduction: (**a**) PI3K/Akt/mTOR; (**b**) AMPK; (**c**) ERK; (**d**) P38, JNK and MAPK signal transduction pathways.

(a)	mTOR Ser^2448^	Akt Ser^473^	Akt Thr^308^	Akt Total
Repressed	[11,44,142]	[11,101,103,142,144,149,150,151,152,153]	[154]	[152,155]
Activated	[38]			
Unchanged/ambiguous		[145]	[145]	
(b)	AMPKα Thr^172^		AMPKβ Ser^108^	
Repressed				
Activated	[113,142,145,146,152,164,166,167]		[152]	
Unchanged/ambiguous				
(c)	ERK Thr^202^/Tyr^204^			
Repressed	[55,103,143,151,174,177,178]			
Activated	[11]			
Unchanged/ambiguous	[176]			
(d)	P38 Thr^180^/Tyr^182^		JNK Thr^183^/Tyr^185^	
Repressed	[101,103,112,174]		[35,103,177]	
Activated	[11,175,176,177]		[11,176]	
Unchanged/ambiguous	[178]			

#### 4.3.7. Effects in Animal Models

We initially retrieved 160 papers studying the effect of cordycepin in diverse animal models. Ten articles were excluded because they did not use purified cordycepin or employed cordycepin derivatives only. Papers studying combination treatment with adenosine deaminase inhibitors were not excluded. Notes were entered from these papers including animal species, type of human diseases models, dose in mg/kg and route of administration. Data were also gathered regarding effects on inflammation, neuronal function and metabolism. While gathering data from papers studying the effect of cordycepin on signal transduction in animal models, we excluded any papers that did not report the precise phosphorylation site (included papers in Table 2). At this stage, the animal models were classified depending on the human diseases and non-diseased models were excluded. The classes were as following depending on the number of publications in each class: cancer (breast cancer, liver cancer, glioma and leukaemia), cardiovascular diseases (dyslipidaemia, cardiac hypertrophy and hypertension), infection, central nervous system disorders (depressive disorders and learning disorders), respiratory diseases (asthma), reproductive disorders, metabolic disorders, bone disorders (osteoporosis and osteoarthritis), endocrine disorders, pain in hyperalgesic priming (a model of transition from acute to chronic pain), inflammation, hepatic diseases, aging (such as age-related sexual dysfunction and oxidative stress), skin disorders and wound healing (Table 3 specifies the allocation). If two relevant classes were identified in one animal model, the classification was assigned to the first field listed. For example, in papers studying the effect of cordycepin on osteoarthritic animal models where pain was also measured, the study was categorised under bone disorders. The range of cordycepin dose administered to the animal was examined in 131 papers, in which the cordycepin dose could be calculated as mg/kg [11,18,23,24,28,37,38,42,64,66,76,79,80,97,107,108,119,120,121,122,126,130,134,142,143,145,150,164,183,184,185,186,187,188,190,191,192,193,194,195,196,197,198,199,202,204,205,206,207,208,209,210,211,212,213,214,215,216,218,219,220,221,222,223,224,225,226,227,228,229,230,231,232,233,234,236,237,293,328,353,354,355].

**Table 2 molecules-26-05886-t002:** Overview of the included publications studying signal transduction pathways in animal models.

	mTOR Ser^2448^	Akt Ser^473^	AMPKα Thr^172^	ERK Thr^202^/Tyr^204^	JNK Thr^183^/Tyr^185^
Repressed	[142,184]	[142]		[184]	[35]
Activated		[183]	[142,164,184]		
Unchanged/Ambiguous					

**Table 3 molecules-26-05886-t003:** Classification of animal models treated with cordycepin. The animal models are classified according to the type of human disease.

Animal Model	Publications
Cancer	[11,18,23,24,64,66,74,79,80,107,108,134,142,143,185,186,187,188,189,190,191,192,193,194,195,196,197,198,199,200,201,202,203]
Cardiovascular	[37,76,120,122,164,183,184,204,206,207,208,209,211]
Infection	[220,221,222,223,224,225,226,227,228,229,230,231]
Central Nervous System	[28,38,116,119,232,233,234,235,236,237]
Respiratory Diseases	[94,117,118,123,124,128,129]
Reproductive Disorders	[244,245,246,247,248,249,250,251]
Metabolic Disorders	[166,238,239,240]
Bone	[59,126,334,342]
Inflammation/Pain	[42,97,110,121,131,133,212,213,214,215,216,217,218,219]

#### 4.3.8. Exclusion of Papers with Image Duplication

During this study, a key paper was retracted because of extensive data duplication (PubMed ID 32764880), and we became aware of the problem of so-called paper mill publications. To try to avoid including any so-far undetected paper mill publications, we examined all articles we had selected for inclusion in the review on duplicated images. No problems on a scale as large as the first paper were encountered, but we found three more papers with clear duplication of one panel. While these may be due to honest mistakes, it shows insufficient care and these papers were removed from consideration. We notified the publishers of our concerns. In two other cases, two images were very similar, but not identical, and we decided to include these but also notify the publishers of these papers.

## Figures and Tables

**Figure 1 molecules-26-05886-f001:**
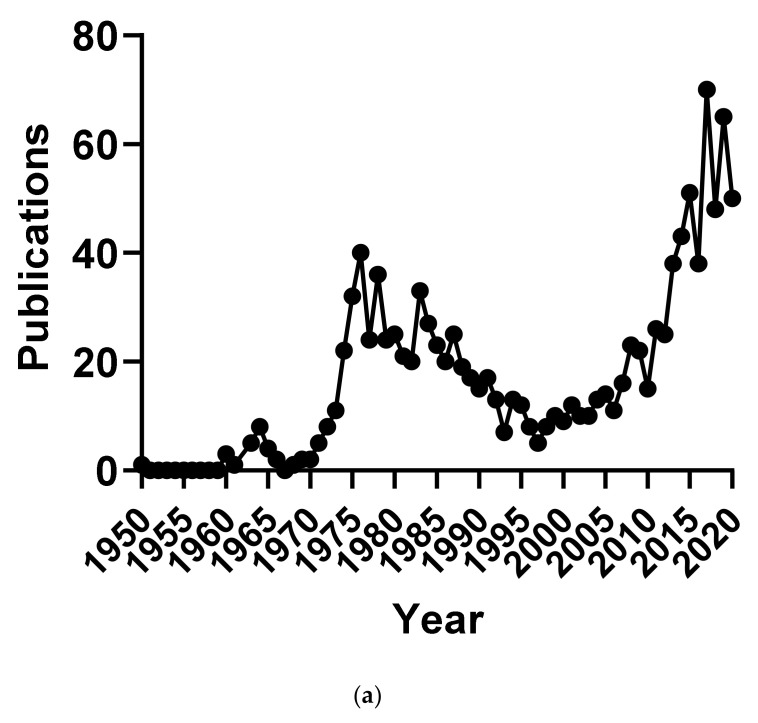
Number of publications on cordycepin by year and geographical origin. (**a**) Number of publications on cordycepin per year of publication 1950–2018; (**b,c**) Geographical distribution of the affiliation of the corresponding authors of publications on cordycepin in two periods: (**b**) 1950–1997 and (**c**) 1998–February 2021.

**Figure 2 molecules-26-05886-f002:**
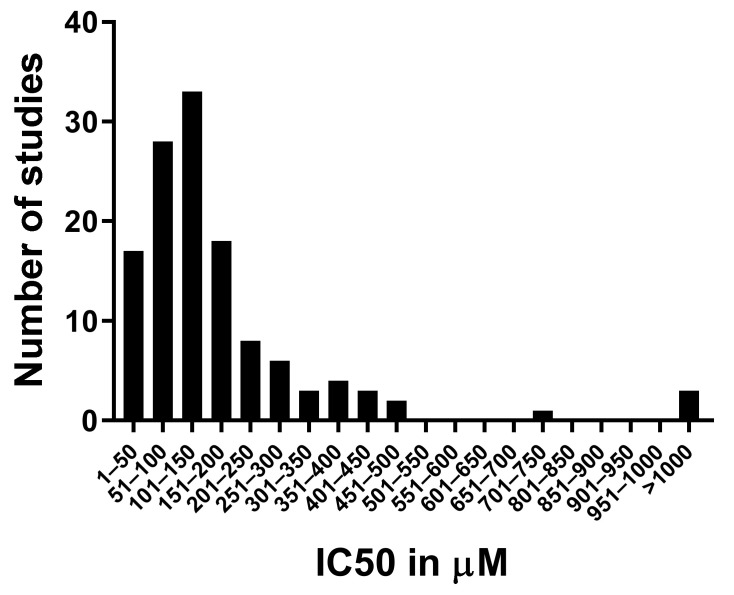
Distribution of the IC_50_ data for cordycepin from the literature. The 50% inhibitory concentration in µM was retrieved from 128 datasets described in 57 papers (listed in the Methods section). The number of datasets with an IC_50_ in each concentration bracket indicated on the Y axis was counted and graphed.

**Figure 3 molecules-26-05886-f003:**
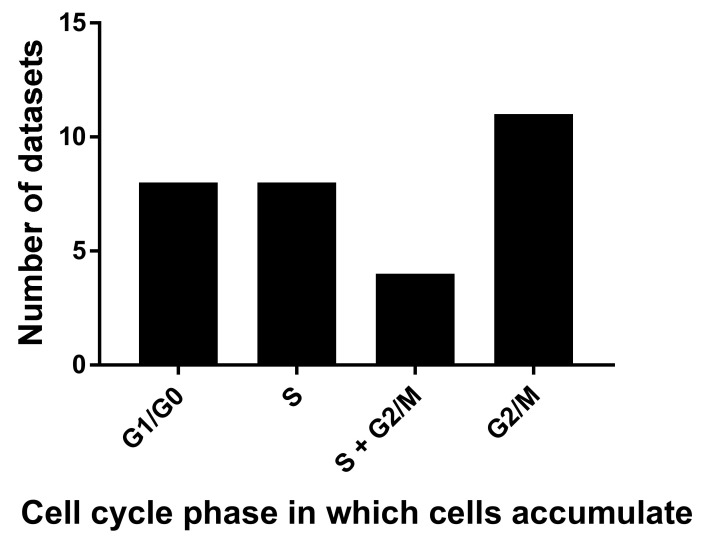
Cordycepin arrests cells in different cell cycle stages. 31 flow cytometry datasets from 18 papers were examined for the cell cycle stage in which cell numbers are significantly increased after cordycepin treatment.

**Figure 4 molecules-26-05886-f004:**
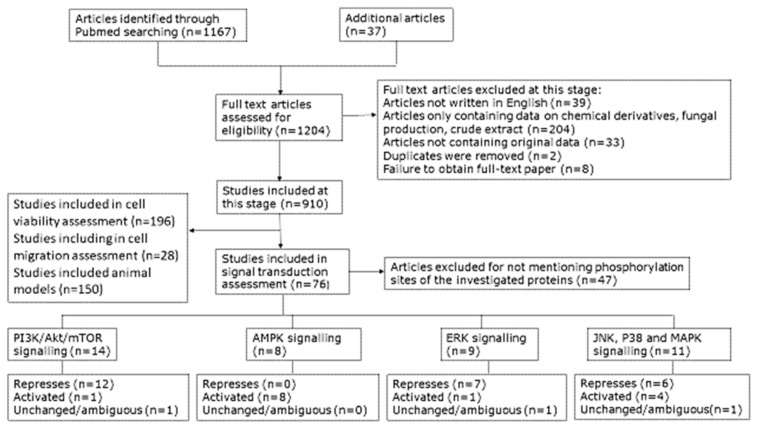
Diagram presenting the selection process for inclusion of publications in this review; n = number of articles.

**Figure 5 molecules-26-05886-f005:**
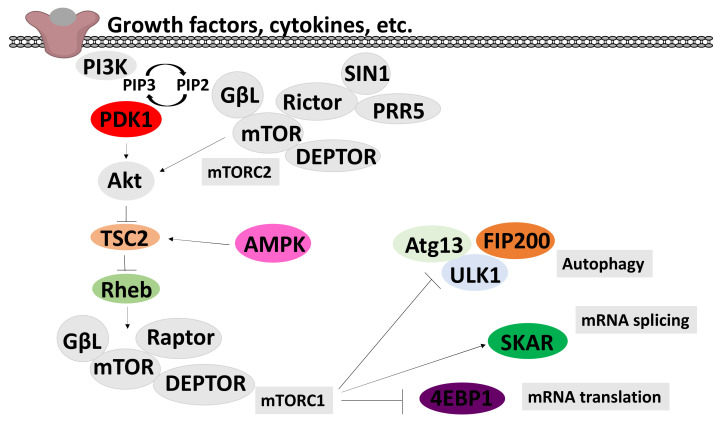
A schematic model of the PI3K/Akt/mTOR signalling pathways. This model integrates PI3K/mTOR/Akt and AMPK signal transduction pathways. Arrows indicate activation and T ends indicate inhibition. Triggered PI3K activates Akt/mTOR cascade, through activation of PDK1 and phosphorylation of AKT in the activation T-loop. mTOR complex 2 (mTORC2), is activated by an as-yet unknown pathway and contributes to the activation of AKT by phosphorylation in the C-terminal. AKT inhibits TSC2, which leads to activation of the small GTPase Rheb and activation of the mTOR complex 2 (mTORC2). The pathway is negatively regulated by AMPK, through activation of TSC2. The PI3K/Akt/mTOR signalling pathway increases protein synthesis through the phosphorylation of the cap-dependent translation inhibitor protein 4EBP1 and inhibits autophagy, leading to the promotion of growth and the anabolic state.

**Figure 6 molecules-26-05886-f006:**
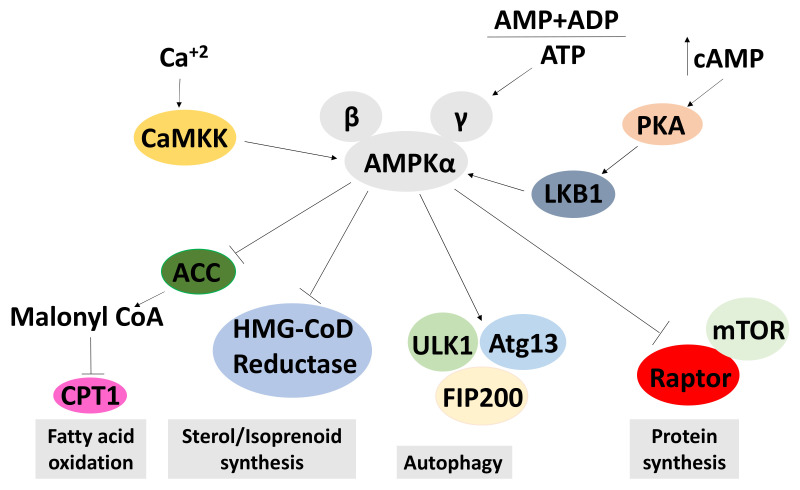
A schematic model of the AMPK signalling pathways. This model shows the activation of AMPK in response to low adenosine triphosphate (ATP) levels, and an increased adenosine diphosphate (ADP) and adenosine monophosphate (AMP). As a result, it activates pathways that produce ATP, thus increasing ATP levels. Conversely, pathways that deplete ATP are repressed by AMPK. AMPK is activated by an increased AMP + ADP to ATP ratio and phosphorylation by CAMKK or LKB1. Activated AMPK inhibits acetyl-CoA carboxylase (ACC), HMG-CoD reductase and mTORC1, leading to an increase of fatty acid oxidation and a reduction in sterol and protein synthesis. Active AMPK inhibits autophagy. An arrow indicates an upregulation of the process and a T end represents a downregulation of the process.

**Figure 7 molecules-26-05886-f007:**
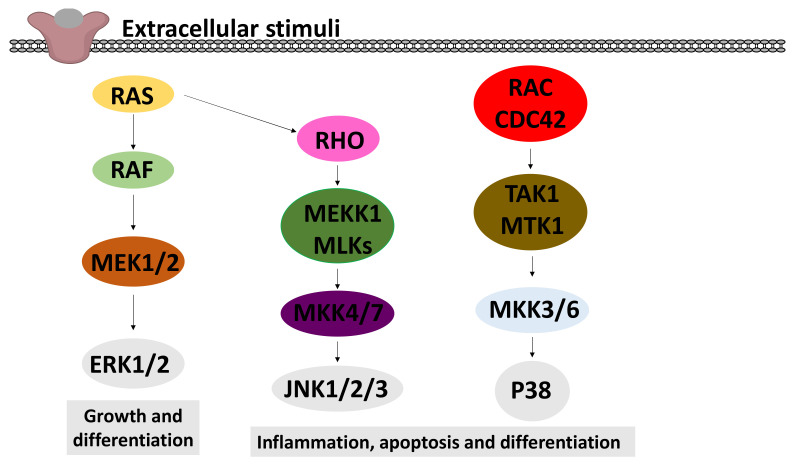
A schematic model of the MAPK signalling pathways. This model shows the three major pathways of MAPK: extracellular signal-regulated kinase (ERK) 1 and 2, c-Jun N-terminal kinases (JNK) 1–3 and p38 MAPK. In all these pathways, the activation of a small GTPase (RAS, RHO or RAC/CDC42) leads to the activation of a MAP kinase kinase kinase (MAPKKK: RAF, MEKK1/MLK, TAK/MTK1), which activates a MAP kinase kinase (MAPKK) by phosphorylation. These MAPKKs finally phosphorylate and activate the MAPKs (ERK1/2, JNK1/2/3 and p38).

**Figure 9 molecules-26-05886-f009:**
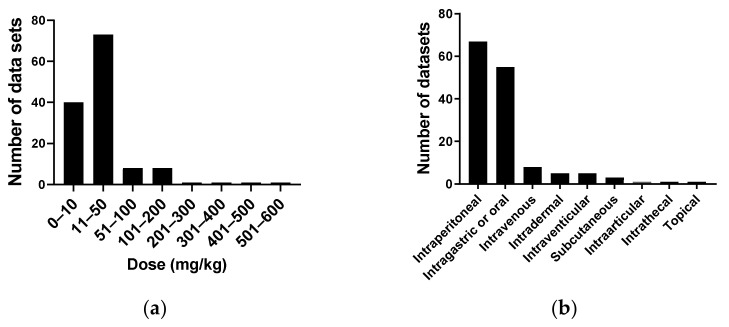
Dose of cordycepin administered to animal models in mg/kg. (**a**) A total of 133 studies were classified according to the range of cordycepin dose administered to the animal model; (**b**) The route of administration of cordycepin in 146 studies.In this systematic review, 167 articles studying the effects of purified cordycepin in a variety of animal models were found (see Table 3). We noted the details of the experimental set-up (species, model of human disease, dose in mg/kg basis and route of administration). The disease models were classified and counted (Figure 10). The majority of studies were of animal models of cancer, closely followed by cardiovascular diseases, infections and central nervous system disorders.

**Figure 10 molecules-26-05886-f010:**
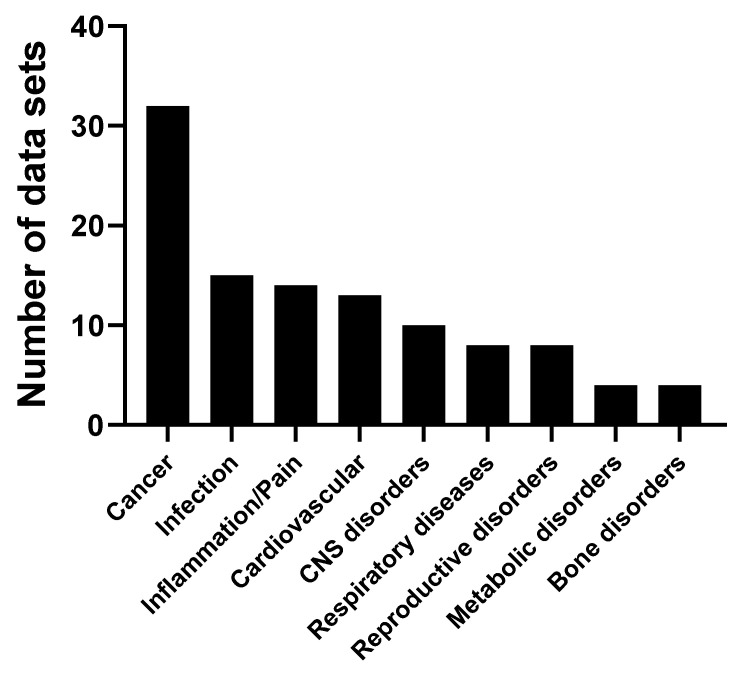
Classification of animal models. A total of 139 articles were classified according to the type of human disease modelled. CNS: central nervous system. Categories which had only one entry were excluded, as described in the Methods section. The exact allocation of papers to disease classes can be found in the Methods section (Table 3).

## Data Availability

All data in the article were extracted from the cited papers as outlined in the Methods section.

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
