# Peer review of "A Systematic Review of the Biological Effects of Cordycepin"

_molecules, 2021, doi:10.3390/molecules26195886_

Round 1

Reviewer 1 Report

The topic of this manuscript is generally well written and very interesting. I have only minor comments that might improve this article. 

  1. Abstract; IC50, 135 µM, This value on cell survival and proliferation is almost the same to that on cell arrest.  The authors refer to cell arrest in line 359, describing that the average is 136 µM. Why don't the authors add the description regarding cell arrest in abstract, such as approximately 135  µM.
  2. Abstract; line 29; secondary responses, This phrase seems to be redundant.
  3. Text; line 276, The authors may use the abbreviation IC50 . The 50% inhibitory concentration (IC50) has already used in line 122. In addition, the abbreviations of several cytokines might be spelled out (ex. lines 155-157). 
  4. Text, need to unify the size of characters 
  5. The title of figures; Overall titles, especially Figs. 1 and 2, should be improved so that readers can understand the figure content easily. 

Author Response

We thank the reviewer for their kind assessment of our manuscript. Here are our responses to the specific remarks:

  1. The quantification of the dose at which effects of cordycepin on the cell cycle are detected is in fact lower than for the IC50 data (the reviewer is mixing up average and median values).  However it is a good point to compare these values in the manuscript and we have now added a sentence to the relevant section of the Results. We have not added the doses at which cell cycle effects occur in the abstract, because the doses are not as carefully titrated as they are for IC50, due to the technical difficulty of doing so. We have also added an explanation to this effect to the relevant section.
  2. We removed this phrase.
  3. We made these modifications
  4. The font changes were implemented.
  5. We clarified the figure titles.

Reviewer 2 Report

Minor remarks

  • The caption of the table should be given above table (Tables 1-3). Each figure should be numbered (a, b, c, ) and that numbering should be explained in the caption of figure.
  • In Figure 1 and Figure 9, the explanation of A and B are given in the caption, but there are not presented in Figure.
  • B, C, E, and F are not presented in Figure 8. Also, it should be presented in the caption of the figure.
  • 50 should be indexed in median inhibitory concentration (IC50). It should be correct in the whole text.
  • All other minor remarks are given in the manuscript.

Major remarks

  • Table 2 should be deleted from Methods and presented in the Results since it is first mentioned there.
  • Table 3 is mentioned in the Methods but was not discussed later. If you do not take into account the results from this table, table should be removed from the manuscript.
  • In the manuscript, the discussion of Figure 8 is first given, and then the discussion of Figure 6. Please, correct it.

Author Response

We thank this reviewer for their helpful comments. All the minor comments were implemented. We changed the mention of figures and tables in the text to match the appearance of the figures and Method tables. The tables are important as the background data for the figures, indicating which papers give which information, but would disrupt the story in the Results. If required, we can place them in supplementary data.